# Nomenclature and Typification of the *Jasione* L. (Campanulaceae) Groups from the Eastern Mediterranean Basin

**DOI:** 10.3390/plants13010050

**Published:** 2023-12-22

**Authors:** Miguel Serrano, Lyuboslava Dimitrova, Santiago Ortiz

**Affiliations:** 1Department of Botany, Faculty of Pharmacy, University of Santiago de Compostela, 15782 Santiago de Compostela, Spain; santiago.ortiz@usc.es; 2Independent Researcher, 3000 Vratsa, Bulgaria; lucy.dimitrova2@gmail.com

**Keywords:** Mediterranean flora, biodiversity, synonymy, taxonomy

## Abstract

In this research, all the published names of *Jasione* (Campanulaceae) in the species, subspecies, and variety ranks for populations from the Balkan and Anatolian peninsulas are reviewed, including names of taxa allied to these groups in surrounding areas. These two areas are among the main centers of diversity of the genus, albeit no recent revisions to address the regional taxonomical complexity of the genus have been published for the Balkans and Anatolia. In this work, the taxonomic status and typification of twenty-six published names are discussed, including the plethora of names within the *Jasione heldreichii*, *J. supina*, and *J. orbiculata* taxonomic complexes. In total, eleven lectotypes and one neotype are designated for names from the aforementioned regions, plus one from southern Italy. This nomenclatural clarification establishes a reference for ongoing and future taxonomic and evolutionary studies of *Jasione* in the Mediterranean basin. In a genus prone to micro-endemism and cryptospeciation, a number of the historically described forms, despite being disregarded by current taxonomic treatments, may, therefore, deserve further attention.

## 1. Introduction

The genus *Jasione* L. (Campanulaceae) is distributed around the Mediterranean basin and throughout Europe, with two centers of diversity at the opposite sides of the Mediterranean Sea, one in the Iberian Peninsula and Morocco and another in the Balkan and Anatolian peninsulas, consistent with the West–East Mediterranean disjunction biogeographical pattern [1,2]. The genus occurs in a number of ecological conditions and geological substrates across a wide elevational range, including coastal and alpine habitats. The combination of environmental niche diversity, complex geological history of the Mediterranean basin and ploidy variation, which includes five ploidy levels [3,4], own data, coupled with relative stasis around a general morphological pattern (i.e., medium-sized or small plants with condensed inflorescences in flower heads surrounded by involucral bracts and small flowers with connate anthers), has produced a large number of polymorphic populations frequently lacking clear morphological distinctiveness. In consequence, many names and different taxonomic ranks have been used for members of the genus (e.g., [4,5,6]). However, most of these names are of unknown evolutionary significance [7], leading to difficulties in species delimitation by morphological approaches [8]. General treatments of the genus thus tend to be overly summarized, with the number of species recognized varying between twelve [9] and seventeen [10]. The evolutionary lineage of *Jasione* seems to be prone to micro-endemism and cryptospeciation, and many of the historically described forms, despite being disregarded by current taxonomic treatments, may, therefore, deserve further attention. Thus, a correct nomenclatural understanding of the clutter of names described so far is particularly important in studies dealing with evolutionary relationships, species delimitation, and biodiversity conservation in such complex groups.

Two important nomenclatural summaries have been provided for the center of diversity of *Jasione* in areas of the western Mediterranean region. Dobignard and Chatelain [6] reviewed the synonym names published for North African populations, while Sales and Hedge [5] tackled the revision of the published names and nomenclatural types of the taxa occurring in the Iberian Peninsula. Parnell [11] had previously addressed the nomenclature of *Jasione montana* L. and other taxa that he considered related, dealing with a huge number of published names associated with the wide range and considerable polymorphism of *J. montana*, which occurs in northwestern Africa and most of Europe (although it is almost absent in the Balkan area). These works were later complemented by nomenclatural precisions in the Iberian–North African *Jasione corymbosa* poir. ex Schult complex [12].

By contrast, there are no recently published revisions for the eastern center of diversity of the genus. The most recent comprehensive work dealing with *Jasione* in the Balkan and Anatolian peninsulas is that of 1926 by Stojanov [13], with a taxonomic treatment recognizing thirteen species and varieties, one from the Italian peninsula. This author addressed the alpine taxa of what he called the *Jasione supina* Sieb. ex Spreng. group, a number of mostly narrow endemic orophytes occurring across different mountain ranges in both peninsulas. This group would include the *Jasione supina* complex, the *Jasione orbiculata* Griseb. ex Velen. complex, and *Jasione bulgarica* Stoj. and Stef. Nevertheless, the evolutionary coherence of this putative group and its internal relationships remain unknown. Stojanov [13] did not consider the *Jasione heldreichii* Boiss. and Orph. complex, the other remaining group occurring in the Balkan and Anatolian peninsulas. Parnell [11] only partly addressed the nomenclature of this complex, describing one variety occurring on the Turkish coasts. A wider study is required that would encompass the nomenclature of the group, within which diverse names with difficult taxonomic interpretation have been described [14].

Tutin [15] subordinated *J. orbiculata* as a subspecies of *Jasione laevis* Lam. in his treatment previous to Flora Europaea [16]. This rather questionable view overshadowed all taxa described or recognized by Stojanov [13] within this species, probably influencing later general treatments that only listed *J. orbiculata* [4,10,17]. The fact that these taxa have the rank of variety has also hampered their modern recognition, although some of them have been included in regional floras (e.g., *J. orbiculata* subsp. *balcanica* Urum. in [18]).

Damboldt [19,20] tackled the taxonomic diversity of the Anatolian *Jasione* orophytes, recognizing two species in this group, i.e., *J. idaea* Stoj. and *J. supina*, the latter of which has four subspecies. He designated nomenclatural types to several names in the Flora of Turkey and the East Aegean Islands [20], although a formal typification of some of them is still lacking.

The research presented here aims to present a nomenclatural revision of the genus *Jasione* in the Balkan and Anatolian peninsulas, the eastern center of diversity of the genus, and complements the current studies of the authors on taxonomy and evolutionary relationships in the genus *Jasione*. The nomenclatural situation of the Balkan and Anatolian taxa of *Jasione*, including the *J. heldreichii*, *J. supina*, and *J. orbiculata* taxonomic complexes, are addressed by summarizing synonyms and designating types. The application of the many names described from the aforementioned regions, plus one from southern Italy, is discussed.

## 2. Materials and Methods

Currently accepted names, including homotypic and heterotypic synonyms, those listed in the Med-Checklist [17], the World Checklist and bibliography of the Campanulaceae [4] and the Euro+Med Plantbase [10] were considered. Other names were obtained from diverse sources (see below), mainly Stojanov [13] and Damboldt [20]. The protologues and original material of names were analyzed in detail, and those names lacking a nomenclatural type were typified in accordance with the International Code of Nomenclature for algae, fungi, and plants [21]. The following herbaria were consulted: B, BR, E, G, FI, K, MPU, P, SO, SOM, GOET, and W. Most of these herbaria conserve the relevant material for the typification of *Jasione* from the studied regions; others are the herbaria of reference of a botanist that worked on *Jasione* in Anatolia and/or the Balkans. The herbaria were consulted in person (as is the case of K and SOM) or digitally, accessed via the JSTOR Global Plants platform, or online virtual herbaria. Herbaria curators were also requested for digitally scanned images from their collections.

## 3. Results and Discussion

### 3.1. Nomenclature and Typification of the Jasione Complexes from the Eastern Mediteranean Basin

All names described for *Jasione* in the Balkan and Anatolian peninsulas at the species, subspecies, and variety ranks were analyzed. Thus, the members of the *J. orbiculata* complex were included, and the representative of the complex in southern Italy is therefore also treated here. The review presents basionyms in alphabetical order, structured as a discussion of each name followed by its synonyms and typification, designating types when required. A list of currently accepted names from relatively recent checklists (e.g., [4]) is provided, and the heterotypic synonyms are indicated (Table 1). The inclusion of a name in the first column of Table 1 only reflects its situation in current general lists, as recognized names may be organized differently after ongoing phylogenetic studies on *Jasione*. Notwithstanding, as a prelude to a more formal taxonomic treatment informed by phylogenetic relationships, Table 1 underlines those names that, in the opinion of the current authors, would merit taxonomic recognition, irrespective of whether future systematic works may propose new combinations of their taxonomic rank.

### 3.2. Jasione bulgarica

Stojanov and Stefanov [22] used material from the Rila mountain range to describe this name, indicating in the protologue a flowering period of July to September 1919 in Rila, but without citing any particular specimen in relation to the name. The protologue includes two illustrations, a photograph with six separated plants (protologue, Image 2 on page 106) that matches several aspects of the description, i.e., glabrous plants with upright stems and creeping roots, and a drawing (protologue, Image 3 on page 107) of a flower showing the character considered most remarkable by the authors, i.e., the free anthers (contrasting with the connate anthers in the remaining species of the genus). The current authors have no evidence of a formal description of the type, and one of the illustrations could, therefore, be selected to designate a lectotype, according to articles 8.1 and 9.12 of the International Code of Nomenclature. Given the importance the name’s authors attributed to the character of the free anthers, the second illustration (protologue, Image 3 on page 107) should be preferred for this purpose. However, several sheets of the material collected by Stojanov and Stefanov in the Rila mountains in July and September 1919 are conserved in herbarium SOM (e.g., SOM 75792, SOM 75780, SOM 75789, among others). It therefore seems reasonable to assume that these authors used at least some of this material to prepare the description, although they did not cite any of these gatherings in the publication. Given that the authors refer to both July and September 1919 in the protologue only as an indication of the flowering phenology in Rila and not as specimen collections, no particular collection can be unambiguously associated with the illustrations. After careful analysis of the sheets in SOM, the current authors could not match any of these specimens to the flowering stems photographed in the first illustration of the protologue. Stojanov and Stefanov [22] stated that they had seen sheets of the new species in the herbaria of V. Stříbrný and I. Urumov, labeled as, and sometimes mixed with, *J. orbiculata*. However, they did not cite any particular gathering that could be chosen as the lectotype. Although none of the specimens conserved in SOM has been explicitly associated with the taxon in [22], they were most probably among the materials used to describe the species as they were collected at the locus classicus in a month (July) indicated in the protologue, and as available specimens according to article 9.4 of the International Code of Nomenclature these specimens could be considered original material. Therefore, and also because a physical specimen will always have more characters than an illustration, the current authors have selected as lectotype the specimen SOM75792! Collected by Boris Stefanoff on 25th July 1919 in Malyovitsa (Мальовица) in the Rila mountain range (Figure 1).

The designated lectotype is, logically, also the type of *Jasionella bulgarica* (Stoj. and Stef.) Stoj and Stef., and it is consequently the species’ type of *Jasionella* Stoj. and Stef (1933), a monotypic genus described on the basis of the free anthers in the flower as the main trait differentiating the genus from *Jasione* L.

***Jasione bulgarica* Stoj. and Stef.**, Österr. Bot. Z. 70: 105 (1921) ≡ *Jasionella bulgarica* (Stoj. and Stef.), Фл. Бълг. изд. 2: 986 (1933)—Lectotype (designated here): Bulgaria, Rila mountains, Malyovitsa. On 25th July 1919. B. Stefanoff s.n. (SOM barcode SOM75792!).

### 3.3. Jasione glabra

Velenovský [23] described this species from material collected in the vicinity of Varna, on the Black Sea coast in Bulgaria, sent to him by A. Javasof. The lectotype is held in PRC (PRC 451934), and it matches the protologue description of a glabrous plant without basal leaves, somewhat undulate stem leaves, and bract teeth and apex apparently not ending in hard awns. This plant can be included in the variability of *J. heldreichii* Boiss and Orph., as considered by Velenovský [24], who later lowered its rank by describing it as *J. heldreichii* var. *microcephala*. This author (*op. cit.*) also wrote that this name alluded to the same plant that he had described under the name *J. glabra*, adding a shorter morphological description, but similar to that included in the protologue of *J. glabra*, giving Varna, Bulgaria as the type locality. According to Art. 6.12 of the International Code, *J. heldreichii* var. *microcephala* Velen. is validated as a replacement name for *J. glabra* Velen., as it is only validated by reference to the earlier name. Thus, it is automatically typified by the type of *J. glabra*. The lectotype was designated by Ančev [18].

***Jasione glabra* Velen.** Oesterr. Bot. Z. 34: 424 (1884) ≡ *Jasione heldreichii* var. *Microcephala Velen.*, Fl. Bulg.: 374 (1891) ≡ *Jasione montana* var. *dentata f. microcephala* (Velen.) Stoj and Stef., Фл. Бълг. Изд, 1,2: 1094 (1925).—Lectotype (designated by Ančev in Fl. Rep. Bulgaria 11: 158. 2012): Bulgaria, Varna, VIII, A. Javasof s.n. (PRC barcode PRC 451934 [photo!]).

### 3.4. Jasione heldreichii

The lectotype [11] shows an annual–biennial habit and bracts with aristate teeth, matching the description in the protologue [25] and the narrower concept used by diverse authors whereby *J. heldreichii* is only considered to be annual–biennial. Two main morpho-ecological patterns can be found within the *J. heldreichii* complex populations (without taking into account some deviant maritime forms): one of annual–biennial hairy plants without sterile rosettes, mostly occurring at mid-low elevations and another of perennial glabrescent plants with sterile rosettes and occurring at higher elevations or northern latitudes [14]. Some authors avoid using the name *J. heldreichii* for the latter forms, preferring either *J. jankae* Neilr. (e.g., [14,26,27]) or *J. dentata* (DC) Halácsy sensu Josifović [28] non sensu orig. for these populations. Nevertheless, current treatments synonymize *J. jankae* under *J. heldreichii*, encompassing both annual–biennial and perennial plants [4,16,17,18,20]. Parnell [11] designated the lectotype of this name by indicating “Macedonia, 1857, Orph. Lectotype: G!” Unaware of this, Hartvig [14] apparently used material from the same collection to designate a lectotype, indicating “Macedonia, Mt. Korthiathi prope Thessalonicam, Orphanides 663 (G-Boiss!)”. Examination of the type material in G clearly shows that the lectotype designated by Parnell [11] belongs to a different collection (Orphanides n° 3130) than that used by Hartvig [14]. The former has the G barcode G60078161. Regarding the variability in *J. heldreichii*, the current authors were able to observe in the type locality that some rare individuals can develop a few sterile rosettes. Indeed, there are some intermediate populations in regard to this trait (own data, personal observations of Miguel Serrano in Greece and Bulgaria), and populations with some individuals developing sterile rosettes and others without them have been recorded after cultivation of the plants and monitoring of their development [29].

***Jasione heldreichii* Boiss. and Orph.**, Diagn. Pl. Orient. Ser. 2, 6: 120 (1859) ≡ *Jasione montana* L. f. *heldreichii* (Boiss. and Orph.) Schmeja, Beih. Bot. Centralbl. 48(2):33 (1931)—Lectotype (designated by Parnell [11]: 265): Macedonia, 1857, Orph. N° 3130 (G barcode G60078161 [photo!]).

### 3.5. Jasione heldreichii var. Papillosa

Parnell [11] used this name to encompass some morphologically deviant populations of group *J. heldreichii* occurring on the Turkish European coast of the Black Sea. These plants are characterized by densely papillose bracts and leaves with thickened margins, and short-triangular calyx teeth. The holotype, held in herbarium E, originates from the T. Uslu n° 2070 collection from Istanbul, Terkos Gölü. The specimen has the E barcode E00913661. Although these populations may simply be coastal ecotypes of *J. heldreichii*, the evolutionary significance must be studied in greater detail as they display some different traits from other dune populations of *J. heldreichii* from the Aegean and Black seas. *J. heldreichii* var. *papillosa* is remarkably distinct from the typical *J. heldreichii*, showing morphological convergence with populations of *J. montaçelena* from sand dune habitats in the western Mediterranean region under similar climatic conditions (i.e., subhumid Mediterranean climate). Indeed, this is the reason for the inclusion of *J. montana* in the Flora of Turkey and the East Aegean Islands [20], an enduring incorrect interpretation of these populations (e.g., [30]).

***Jasione heldreichii* var. *papillosa* J. Parn.**, Watsonia 16 (3): 266 (1987)—Holotype: (Designated by Parnell [11]: 265): Turkey, Istanbul, Terkos Gölü, Uslu n° 2070 (E barcode E00913661 [photo!]).

### 3.6. Jasione idaea

This name was published by Stojanov [13] to describe a narrow endemic species in the Kazdağı, the ancient Mount Ida, in northwestern Anatolia, using one collection of Sintenis (Iter trojanum, 1 Aug. 1993, n° 532). Damboldt [20] addressed the typification of this name from this collection. However, he designated a holotype by indicating “holo. W! iso. B, M, E!, G! GOET, K! LD”. Many specimens exist from the original collection, and the lectotype should be designated in accordance with article X.X of the International Code. Therefore, the current authors designated one of the many specimens from the original collection as the lectotype, the sheet in herbarium E with barcode E00913653, which is shown in Figure 2.

***Jasione idaea* Stoj.** In Notizbl. Bot. gart. Berlin 9: 556 (1926)—Lectotype (designated here): Turkey, Kazdağı (ancient Mount Ida) 1 August 1883, Sintenis n°532, iter Trojanum (E barcode E00913653 [photo!]).

### 3.7. Jasione jankae

This name is listed as a synonym of *J. heldreichii* Orph. ex Boiss. in Flora Europaea [16], the Med-Checklist [17], and in the World Checklist of the Campanulaceae [4], although it is recognized in the Euro+Med PlantBase [10], as well as in different regional floras from the Balkan Peninsula (e.g., [14,24,26]), while other floras consider it a synonym of *J. heldreichii* (e.g., [18,31,32,33]). Neilreich [34] described this species from material collected in the rocky areas of Treskovac, NW of Svinica, in Romania (current names in Romanian language, Trescovăţ and Sviniţa). In herbarium C (C barcode C10009017) there is a gathering collected by Viktor Janka in “rupestribus m. Treszkovácz inter Drenkova et Svinieza ad Danubium inferioren Banatus”, on 4 July 1870 during his Iter Banaticum. The specimen is identified in the sheet as *J. jankae* Neilr. and two manuscript labels dated September 2004 are glued on it (Figure 3). One label, by I.C. Hedge, indicates that “the formal lectotypification of *J. jankae* will be published in due course”, and the other, by S. Prince, indicates “Lectotype of *Jasione jankae* Neil. = *J. heldreichii* Boiss & Orph.” The current authors have no evidence of any subsequent formal publication and, therefore, have designated this specimen as the lectotype. There is no doubt that this material was used by Neilreich to describe the species, as correctly pointed out by I.C. Hedge and S. Prince, as the label clearly conveys the location cited in the protologue, and the specimen matches all points in Neilreich’s description.

The status of *J. jankae* in relation to *J. heldreichii* has been controversial, even in some of the treatments that recognize it as different from *J. heldreichii* [14]. Ciocârlan [27] addressed the question by adopting the position of differentiated species but provided little more evidence than the differing morphological features already described in the protologue. Thus, *J. jankae* may refer to plants similar to *J. heldreichii* but differing in the perennial habit with sterile rosettes, rather than to annual–biennial habit without rosettes in *J. heldreichii*, dentate bracts but with teeth half the width of the bract teeth rather than pinnate-pectinate bracts with aristate apex. However, as intermediate individuals are found, a wider concept of *J. heldreichii*, including plants from mountain or northern areas where sterile rosettes are more frequent, may appear to be more consistent with the current state of knowledge. Nevertheless, more detailed studies are needed to ascertain whether the observed variation can be linked to independent evolutionary lines. Different degrees of introgression in *J. heldreichii* populations from current or past sympatric populations of representatives of the *J. orbiculata* and *J. supina* perennial groups should not be ruled out as a source of morphological and ecological differentiation within the *J. heldreichii* complex.

***Jasione jankae* Neilr.** Diagn. Aufz. Ungarn. Slavon, Gefässpfl., Nachtr.: 43 (1870) ≡ *J. heldreichii* Janka ex Nyman, Consp. Fl. Eur. 2: 486 (1879) ≡ *J. montana* var. *jankae* (Neilr.) Stoj. and Stef., Фл. Бълг. изд, 1,2: 1094 (1925) ≡ *J. montana* f. *jankae* (Neilr.) Schmeja in Beih. Bot. Centralbl. 48: 33 (1931)—Lectotype (designated here): Romania, Sviniţa, Trescovat peak. On 4th July 1870, V. Janka s.n. (C barcode C10009017 [photo!]).

### 3.8. Jasione montana var. Dentata

De Candolle [35] described this name from plants collected by P.M.R. Aucher-Eloy in the “Olympo Bithino”, i.e., in the UluDag, Turkey. The G-De Candolle herbarium (GDC) holds a gathering collected by Aucher-Eloy from the type location in 1837, number 1880 (G barcode G00329536), matching the morphology described in the protologue (i.e., spiny-dentate acute bracts) shown here in Figure 4. The current authors are not aware of any formal description of the type of this name and have, therefore, designated this specimen as the lectotype. Another sheet from the Aucher-Eloy n° 1880 collection in K (K barcode K000781177) would be an isolectotype. The name and the more frequently recognized Halacsy’s combination [36] is *Jasione dentata* (DC.) Halacsy has been used for plants from Anatolia and from the Balkan Peninsula (e.g., [28,36]). Although the most commonly followed current treatments consider this name a synonym of *J. heldreichii* [16,18], more detailed evolutionary studies are required to clarify this point, given the complexity of the group.

***Jasione montana* var. *dentata* DC**. in Prodr. VII.:415 (1839) ≡ *Jasione dentata* (DC.) Halacsy, Consp. Fl. Gre. 2: 280 (1902)—Lectotype (designated here): Turkey, Bursa province, Uludağ (Olympo Bithino), 1837, P.M.R. Aucher-Eloy 1880 (G barcode G00329536 [photo!]).

### 3.9. Jasione orbiculata

This name has been generally recognized at the species rank [4,17,18], despite its treatment in Flora Europaea subordinated as a subspecies of *J. laevis* Lam. [16], which has influenced some subsequent works [14,37]. Grisebach [38] created this name in the entry for *J. supina* Sieb. in his Spicilegium florae Rumelicae et Bithynicae, although without indicating the morphological features that characterize it and differentiate it from *J. supina*, and without associating this name with any particular location. In the entry he cited several Balkan and Anatolian regions and localities (Macedoniae, Bithyniae, Kobelitza, Peristeri, Olympo in Turkey) but without connecting the new name to any of them. Velenovský [24] validly described the species, separating the Balkan plant from *J. supina*, recognizing the authorship by Grisebach and explicitly referring to the name proposed in the Spicilegium. Velenovský himself [24] did not see the original Balkan material from Peristeri and Kobelitza cited by Grisebach [38]. Stojanov [13] also did not cite this material despite carrying out a huge revision of specimens of the group in different European herbaria. Strid [39] found, in this author’s words, a “rather fragmentary specimen” at G-BOIS, labeled “*Jasione orbiculata* Gr. Scardus. Turquie d’Europe. Grisebach 1842”, which was designated as a lectotype. Scardus refers to the Šar Mountains, the mountain range to which Kobelica [Kobelitza] peak belongs. 1842 is presumably the year the specimen was received in Geneva, as it is known that Grisebach collected in Kobelica mountain on 21st July 1839 [39]. Despite having consulted diverse herbaria, including GOET -the herbarium of reference of Grisebach [40]—it was not possible to locate any other material collected by Grisebach that could be related to this taxon.

***Jasione orbiculata* Griseb. ex Velen.**, Fl. Bulg.: 375 (1891) ≡ *J. laevis* subs. *orbiculata* (Griseb. ex Vel) Tutin, in Bot. J. Linn. Soc. 67: 277 (1973). Lectotype: (Designated by Strid [39]: 280): Republic of North Macedonia, Šar Mountains, Kobelica. G-BOIS.

### 3.10. Jasione orbiculata var. Balcanica

This name has been considered by Lammers [4] as a synonym of *J. orbiculata* s. str. and is not listed in general treatments (e.g., [10,17]), although it has been recognized by Ančev [18]. Urumov [41] described this name, indicating two localities and a year “In graminosis alpinis m. Jumruk-Cal, Mara-Gidik legit a. 1897”, with the former location being in the Rila Mountain range and the latter in the Central Balkan Mountain range. The current authors have found a gathering collected by Urumov in 1897 in Jumruk-Cal in the SOM herbarium (SOM75664) that matches the description in the protologue, i.e., leafed ascendent stems, undulate–denticulate obtuse leaves, wide, and obovate bracts overpassing the flowers and subsessile flowers (Figure 5). The current authors have not found any evidence of formal typification of this name and have, therefore, designated this specimen as the lectotype.

***Jasione orbiculata* var. *balcanica* Urum.**, Österr. Bot. Z. 49: 56 (1899)—Lectotype (designated here): Bulgaria, Rila mountain range, Jumruk-Cal, 1897, I.K. Urumov s.n. (SOM75664!).

### 3.11. Jasione orbiculata var. Bosniaca

Like all the varieties of *J. orbiculata* described by Stojanov [13], this is listed as a synonym of *J. orbiculata* s. str. By Lammers [4]. Although this plant exhibits a markedly different morphology from the typical *J. orbiculata*, the use of this name is virtually absent in the regional botanical literature [42,43,44]. Stojanov [13] used diverse collections from a wide range to describe this name, including gatherings from NW Macedonia, Albania, Montenegro, and Bosnia-Hercegovina. Among the diverse materials he cited in the protologue, there is one gathering collected by E. Brandis in Bosnia from Vranika Sekira peak, 15 July 1892. There is one specimen (MPU barcode MPU016152) from this gathering in MPU. The sheet has a glued label, dated 2010 by P.R. Fabre, indicating “isosyntype of *J. orbiculata* var. *bosniaca*” (Figure 6). The current authors have no evidence of a formal typification of this name and have, therefore, designated this specimen as the lectotype. This gathering was used by Freyn to designate the name “*Jasione propullulans*” [45], which was invalidly published (see below).

***Jasione orbiculata* var. *bosniaca*** Stoj., Notizbl. Bot. Gart. Berlin-Dahlem 9; 553 (1926)—Lectotype (designated here): Bosnia, Vranika Sekira peak, 1892, E. Brandis 2866. (MPU barcode MPU016152 [photo!]).

### 3.12. Jasione orbiculata var. Italica

This name is considered a synonym of *J. orbiculata* s. str. by Lammers [4] and is not included in the Checklist of the Italian vascular flora [46] or used in recent works addressing the conservation status of its populations [47], although it is recognized in Pignattti’s Italian floras [48,49]. This name refers to a narrow endemic restricted to the Sirino Massif, Basilicata, southern Italy [50]. Stojanov [13] cited it in three gatherings included in the protologue, one from “Montis Serino del Papa”, 14 July 1877 collected by Huter, Porta, and Rigo, another from “Monte Serino, pr. Lago Negro”, July 1877 collected by Huter, and another from “M. Serino” July 1898 collected by G. Rigo n°493 during his fourth Iter italicum. The current author has found sheets of the first gathering in the FI herbarium (FI barcodes FI064103, FI064104, FI064105 and FI064106) and P herbarium (P00270615, P00270751, P00270616) and from the third gathering in the E herbarium (E barcode 00913671) and P (P00270620, P00270621, P00279679); all are identified as *J. supina* by the collectors. The current authors have not found any evidence of formal typification of this name and have, therefore, designated sheet FI064103 as the lectotype (Figure 7). The several individuals on this sheet match the protologue description of prostrate stems and large, wide bracts, a trait that led Stojanov to consider this variety akin to *J. orbiculata* var. *bosniaca*, a taxon with erect stems. Therefore, the sheet captures the variability of the taxon, including the abundance of rosettes on long creeping stems and the absence or scarcity of true stolons.

***Jasione orbiculata* var. *italica* Stoj.**, Notizbl. Bot. Gart. Berlin-Dahlem 9; 554 (1926)—Lectotype (designated here): Italy, Basilicata, Massicio del Sirino, Monte Papa (in pascuis montis Serino, del Papa 2.000 m) 14 July 1977, Huter, Porta and Rigo, n° 473 (FI barcode FI064103 [photo!]).

### 3.13. Jasione orbiculata var. Supinoides

This name is considered a synonym of *J. orbiculata* s. str. by Lammers [4] and is not recognized in regional floras or taxonomic lists [14,51]. Stojanov [13] described the name from material collected in the Kaimaktsalan massif, on the border between North Macedonia and Greece, and from Smolikas peak in the Pindos Mountain range in Epirus, Greece. However, as noted by Hartvig [14], there is significant variation among the Greek populations, with mat-forming plants with creeping, rooting stems and ovate-rhomboidal bracts in the SW populations and more cespitose plants with narrower lanceolate-linear bracts in the northernmost Greek populations. While the epithet “supinoides” (i.e., resembling *J. supina* Sieb., a cespitose plant) would be suitable for populations from the mountains on the border between North Macedonia and Greece, the description in the protologue could encompass all of these Greek forms. However, the mention of oblongo-rhomboidea bracts is more consistent with the plants from the Pindus Mountain range. The current authors have located two sheets in herbarium P (P barcodes P00270608 and P00270609) of a collection of Antonio Baldacci from Smolikas, one of the three gatherings cited by Stojanov in the protologue. The current authors have not found any evidence of previous typification of this name and have, therefore, designated one of these sheets as the lectotype (Figure 8).

***Jasione orbiculata* var. *supinoides* Stoj.**, Notizbl. Bot. Gart. Berlin-Dahlem 9; 554 (1926)—Lectotype (designated here): Greece, Epirus, mount Smolika, supra Kerasovo. 18 July 1896, A. Baldacci n° 247 (P barcode P00270608 [photo!]).

### 3.14. Jasione orbiculata var. Orbelica

Velenovský [52] described this name in supplementum I of Flora Bulgarica. In this work, he tackled the morphological description of two varieties of *J. orbiculata* Griseb., conveying doubts about which was Grisebach’s plant. He left the first variety unnamed, supposedly because he found it was more similar to the typical *J. orbiculata*, although this is not certain. Secondly, the var. *orbelica* was described as a plant with longer, thicker, and straight stems, larger denticulate leaves and glabrous acutely denticulate bracts. Stojanov and Stefanov [22] and Stojanov [13] noted that the morphological description of this name matched *J. bulgarica*. However, as Velenovský [52] said nothing about one of the main features of *J. bulgarica*, i.e., the free anthers, the description could also match some forms of *J. orbiculata*. Stojanov [13] thought that this was the reason for Velenovský’s doubts about which of the two varieties that he morphologically described would correspond to Grisebach’s *J. orbiculata*. Nevertheless, the forms of *J. orbiculata* that could match Velenovský’s description do not occur in Bulgaria, where populations that Stojanov [13] called *J. orbiculata* var. *bosniaca* are found. Velenovský [52] cited material collected by Stribrný in Gjumrukcal and Musala peaks, by Reiser in the Demir Kapija-Balkan peak, and by Urumov in the Trojan-Balkan Mountain range, without linking these localities to any of the varieties. The current authors have located two gatherings from Velenovský’s collection in the PRC herbarium, with manuscript labels by this author indicating “*J. orbiculata* var. *orbelica*”: one collected by Stribrný in 1897 in Musala (PRC code PRC451250) and the other collected in 1893 by Reiser in Demir Kapija (PRC code PRC451251). The latter only includes plants matching the description in the protologue, while the former includes two different types of plants, with one at the bottom that does not match the description but corresponds to *J. orbiculata* var. *balcanica* Urum., and five stems in the upper part of the sheet that match the description of var. *orbelica*. To avoid any ambiguity, and in the absence of any evidence of previous formal typification of the variety, the current authors have designated Reiser’s sheet annotated by Velenovský as the lectotype (Figure 9). In any case, this name is a synonym of *J. bulgarica*.

***Jasione orbiculata* var. *Orbelica* Velen**., Fl. Bulg. Suppl. 1: 188 (1898)—Lectotype (designated here): Bulgaria, Demir Kapija peak (Pirin Mountain range), 1893, Reiser s.n. (PRC barcode PRC451251 [photo!]).

### 3.15. Jasione propullulans

Freyn and Brandis [45] equated this name to *Jasione orbicularis* (sic) Griseb., evidently meaning *J. orbiculata* Griseb. The species is not recognized in Flora Europaea [16], the Med-Checklist [17], or the World Checklist of the Campanulaceae [4] and is not even listed as a synonym. However, it is considered a validly published synonym of *J. supina* Sieb. Subsp. *supina* in Plants of the World Online (POWO, 2023). The description indicates that the plant is from Vranica Planina, in Bosnia. As *J. supina* subsp. *supina* is a plant from eastern Anatolia; the synonym status in POWO (2023) can be ruled out. The origin of this misinterpretation seems to be based on a comment in the protologue stating that this plant is the same as that collected under the name *J. supina* by Velenovský in 1887 in western Bulgaria. This name could have been given greater consideration if it had been validly published, in which case the populations of the taxon known as *J. orbiculata* var. *bosniaca* Stoj. Would have been given a higher taxonomic rank. However, the indication by Freyn and Brandis [45] does not include any morphological description, and therefore it was not correctly published according to the International Code of Nomenclature.

**Nom. Nud. *Jasione propullulans*** in Freyn Verh. K. K. Zool.-Bot. Ges. Wien 38: 618 (1888).

### 3.16. Jasione supina

This name was published by Sprengel [53], who recognized Sieber as the author’s name and vaguely indicated “Asia minor?” as the type locality. Damboldt [19] revised the names of the *J. supina* group, adopting De Candolle’s [35] concept of the species for use in the Flora of Turkey and the East Aegean Islands [20]. This interpretation of *J. supina* is based on the population collected by Aucher (n° 1881) in 1837 from the “Bithynian Olympus”, i.e., the Uludağ mountain in Bursa province, Turkey. Neither Damboldt nor the current authors have located any original material collected by Sieber, although it may exist somewhere. Damboldt [20] addressed the typification of this name with Aucher’s n° 1881 gathering, indicating “Neotype: Turkey, Bursa, Olympus Bithynus (Ulu Da.), 1837, Aucher 1881 G! W!)”. Given that two sheets in different herbaria were designated as neotypes, there is some ambiguity regarding the actual type. Therefore, according to the International Code (Art. 9.17), as a second-step typification [21,54], the current authors have designated the sheet in G (G barcode G00329587) as neotype (Figure 10).

***Jasione supina* Sieber ex Spreng.**, Syst. Veg. I:1810 (1825) emend. DC., Prodr. 7:416 (1839). Neotype (designated here): Turkey, Bithynian Olympus (Uludağ mountains), 1837 Aucher (n° 1881) (G barcode G00329587 [photo!]).

### 3.17. Jasione supina subsp. Akmanii

This name and rank are recognized by Greuter et al. [17] and Lammers [4], although the close affinity to *J. supina* subsp. *pontica* (Boiss.) Damboldt has been noted since the inception of this name [19]. This plant was described as a narrow endemic from the volcanic mountain range of Köroglu in Bolu province [19,20], differing from subsp. *pontica* only by the entire bracts instead of dentate bracts. Damboldt only had access to one collection (Akman n° 3051, 14 VII 1975) and in the protologue indicated (holo. ANK), while in Flora of Turkey [20] wrote “Type (holo E! iso. ANK). The current authors have found one specimen in E (E barcode E00913665) identified as type by a manuscript label by Damboldt. This may imply that the only indication of ANK for the type in the protologue is a typographical error that deleted the reference to “E!”. If this were the case, the fact that at least two specimens from the original material collected by Akman exist (in E and ANK) and the unfortunate putative misprint designating as holotype some material explicitly not seen by Damboldt, and therefore, not used by him to prepare the protologue, would leave this name without proper typification. Nevertheless, the absence of the “!” after ANK [19,20], indicating that this material was unseen by this author, could also be attributed to a typographical error. For the sake of precaution, the current authors will be conservative and consider that Damboldt properly designated in the protologue a holotype existing in the herbarium ANK. Nevertheless, if after revision of the material conserved in ANK, no indication by Damboldt in the material from the Akman n° 3051 collection is found, the current authors consider that a lectotype should be designated, preferably the specimen in herbarium E.

***Jasione supina* subsp. *akmanii* Damboldt**, Notes R. B. G. Edinb. 35: 51 (1976)—Holotype: Turkey, Bolu: summit of the Köroglu Da., volcanic rock, 2000–2400 m, 24 VIII 1975, Akman 3051 (ANK).

### 3.18. Jasione supina var. Pontica

Boissier [55] described this name using two collections from the mountains of northern Anatolia, one from “Ponti Armeni ad Aliskeri Khan, Kotschy n° 208” and another from “Bousdouandagh Ponti Lazici supra Khabackar Balansa s.n.” This taxon has been treated as a species [13,56] and a subspecies [19,20], with the latter treatment being followed by Greuter et al. [17] and Lammers [4].

Damboldt [20] addressed the typification of the name, indicating “Type: Trabzon, in pascuis alpinis inter Tschayrlar (Çayirlar) et Alischeri Chan, 28 VII 1859, Kotschy 208 (holo, G! iso. W!)”. Given that Boissier included different collections and specimens in the protologue, the formal typification of the name cannot be achieved until a specific lectotype is designated. The current authors have found specimens from the original Kotschy’s collection in herbarium G (G barcode G00781703) and from Balansa’s collection (G barcode G00781702). Both match the protologue description of a glabrous plant with smaller but wider calyx lobes than *J. supina* s. str. Damboldt’s first-step typification using the gathering of Kotschy should be specified by second-step typification [54], and, therefore, the current authors have designated one of Kotschy’s specimens as a lectotype (Figure 11).

***Jasione supina* var. *pontica* Boiss.**, Fl. Or. 3:886 (1875) ≡ *J. pontica* (Boiss.) Hand-Mazz. in Ann. Nat. Hofmus. Wien 23:192 (1909). ≡ *J. supina* subsp. *pontica* (Boiss.) Damboldt in Notes R. B. G. Edinb. 35: 51 (1976)—Lectotype (designated here): Turkey, Trabzon, inter Tschayrlar (Çayirlar) et Alischeri Chan 28 VII 1859, Kotschy 208, (G barcode G00781703 [photo!]).

### 3.19. Jasione tmolea

This taxon is recognized at the subspecies rank subordinated to *J. supina* by Damboldt [20], Greuter et al. [17], and Lammers [4], referring to a narrow endemic from Mount Bozdagh, the ancient Mount Tmolus, in Izmir province, Turkey. Stojanov [13] described this name citing three different collections, one by Balansa (n° 333 July 1854, Sommet du Tmolus Occidentalis, au dessus de Y’aila de Bozdagh), one by Boissier (s.n., 1866 Smyra: Tmolus) and one by Frivaldsky (s.n. 1845, Smyrna, sub *Phyteuma*). Damboldt [20], in the Flora of Turkey and the East Aegean Islands, listed these three collections and indicated them as “syntypes” in the G and W herbaria. The current authors have found several specimens of the Balansa 333 collection in P (P barcodes P00270579, P00270580, and P00270578) and GOET (GOET barcode GOET005895). These specimens match the description in the protologue, and in the absence of any evidence of a previous formal typification of this name, the current authors have designated one of these specimens as the lectotype (Figure 12).

***Jasione tmolea* Stoj.**, Notizbl. Bot. Gart. Berlin-Dahlem 9; 550 (1926) ≡ *Jasione supina* subsp. *tmolea* (Stoj.) Damboldt in Notes R. B. G. Edinb. 35: 51 (1976)—Lectotype (designated here): Turkey, Izmir, Bozdağ (Tmolus) Balansa n° 333, July 1854 (P barcode P00270580 [photo!]).

Finally, there remain two names for which the current authors were not able to locate any specimens from the original collections and which remain without any formal typification for the moment. Both names are included in the varietal rank and are synonyms of higher-rank names. In both cases, they refer to rare deviant morphs co-existing in populations among the typical forms:

***Jasione supina* var. *hirtula* Stoj**., Notizbl. Bot. Gart. Berlin 9: 550 (1926), Type collection: Olympus Bythinicus (Boissier 1842, in herbario Musei Vindobonensis). A synonym of *J. supina* subsp. *Supina*, defined by its hairier habit.

***Jasione pontica* var. *microcephala*** Freyn in Bornmüller, Plantae Anatoliae orientalis 1910, 2195 pro forma. Type collection: Paphlagonia, in monte Ilghaz-Dagh alt. 2000 m. 12-VIII-1890 Leg. J. Bornmüller. A synonym of *J. supina* subsp. *Pontica*, defined by its smaller flower heads.

## Figures and Tables

**Figure 1 plants-13-00050-f001:**
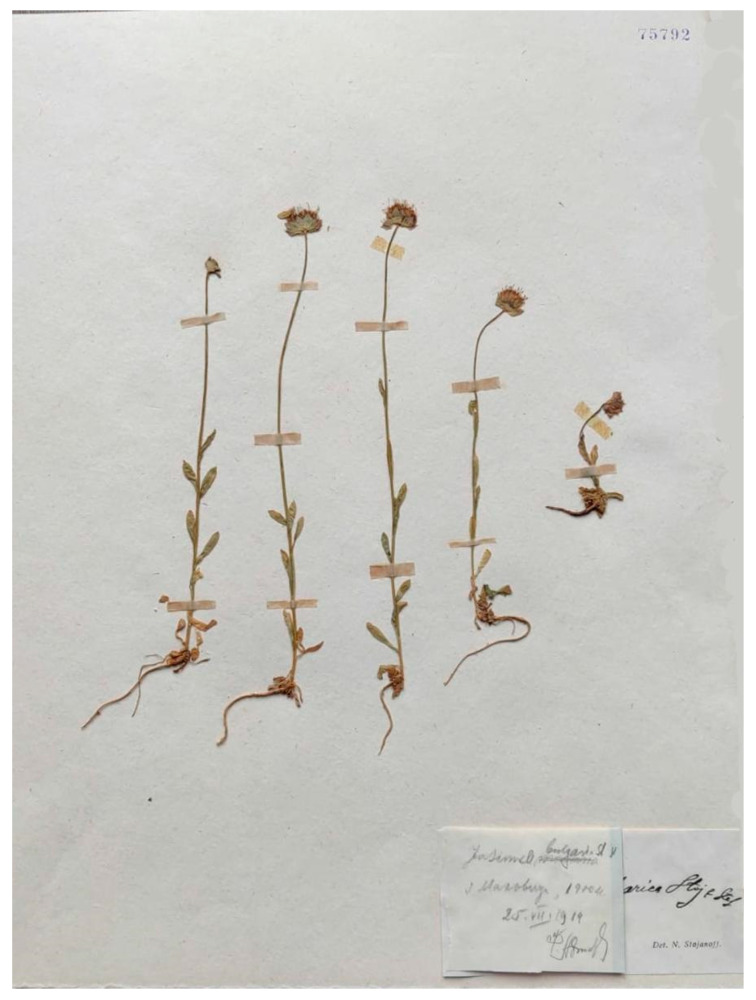
Lectotype of *Jasione bulgarica*, designated here (SOM75792). The handwritten label reads “*Jasione bulgarica*, Malyovitsa, 25.07.1919, B. Stefanoff”.

**Figure 2 plants-13-00050-f002:**
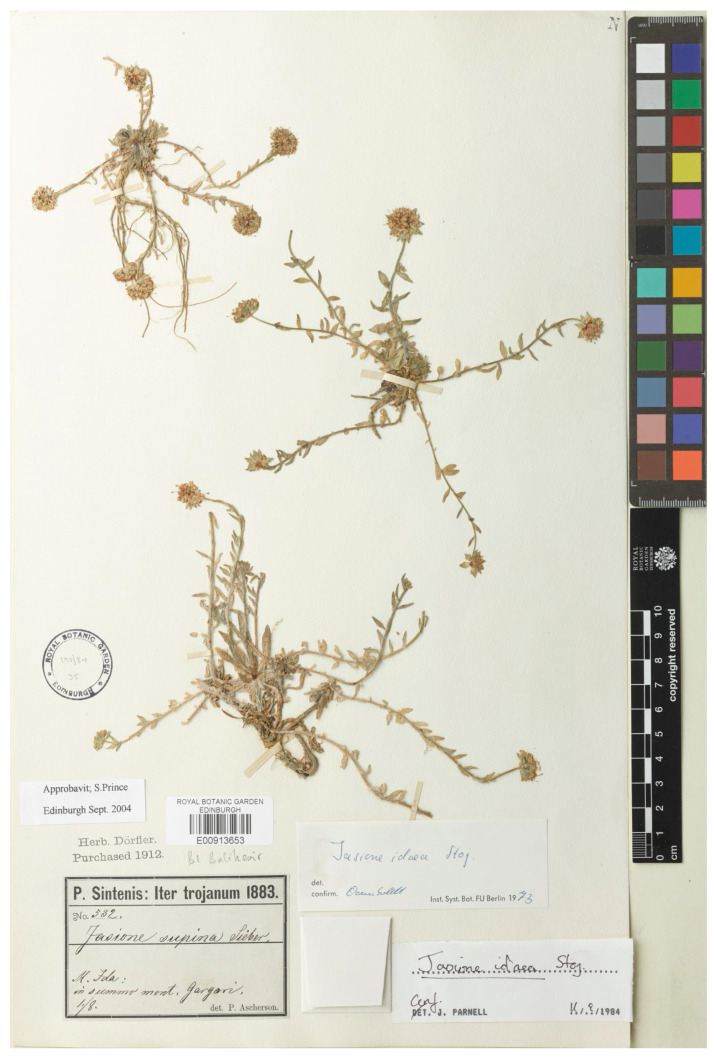
Lectotype of *Jasione idaea*, designated here (E00913653).

**Figure 3 plants-13-00050-f003:**
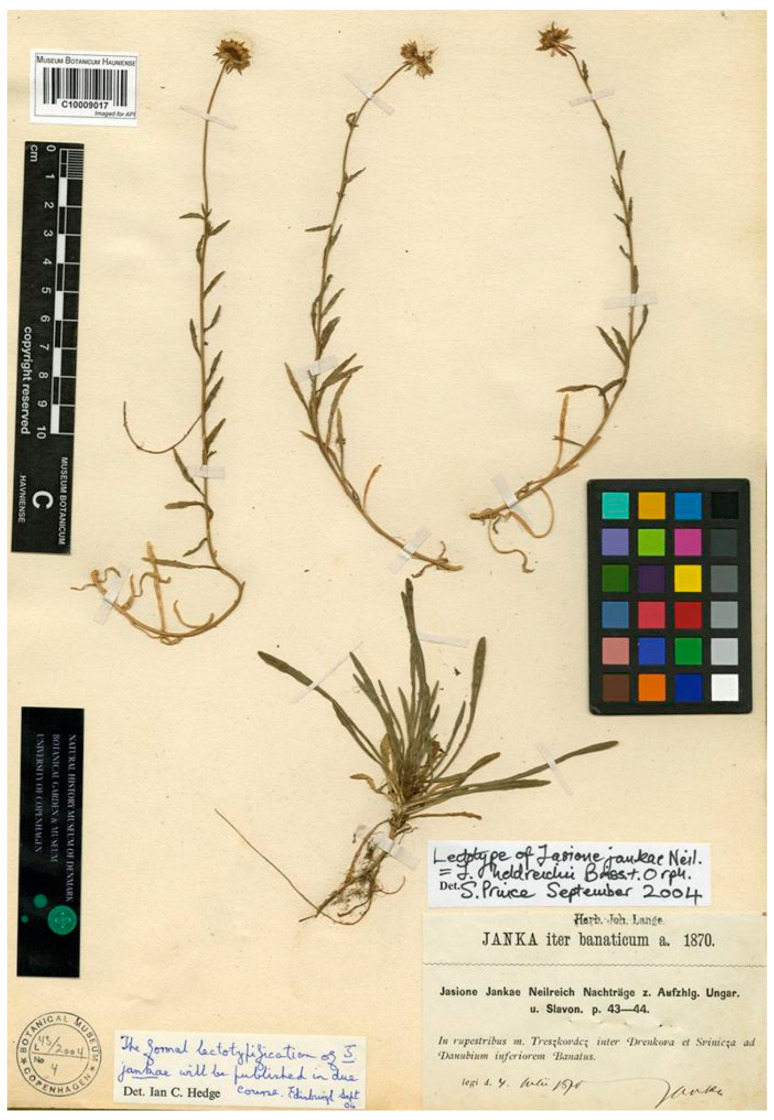
Lectotype *of Jasione jankae*, designated here (C10009017).

**Figure 4 plants-13-00050-f004:**
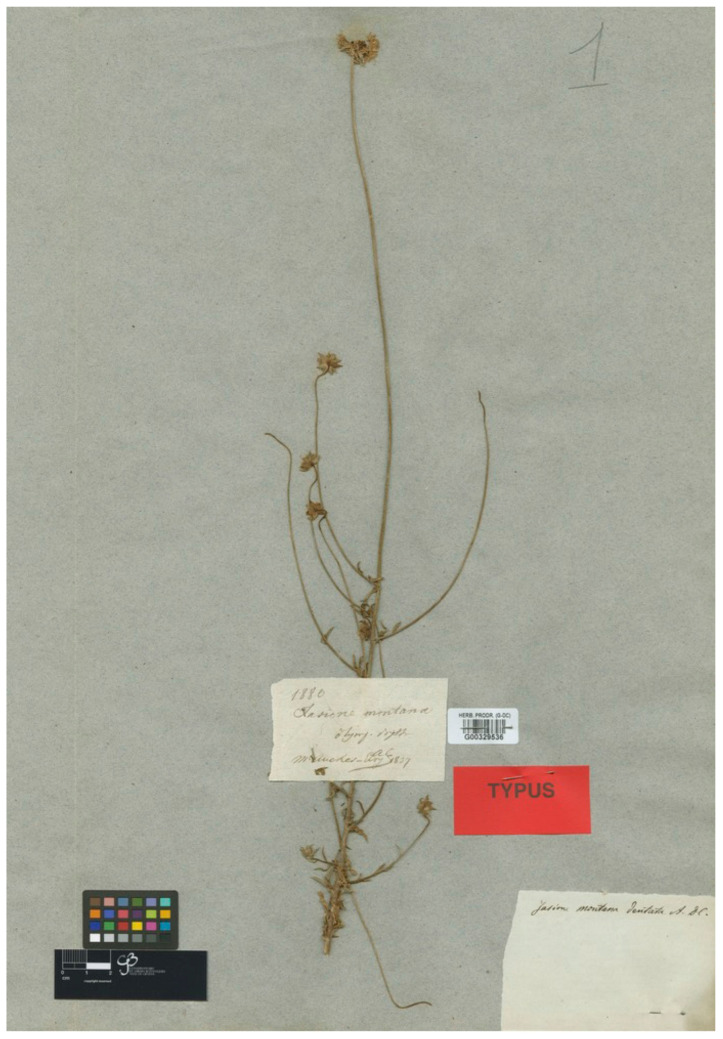
Lectotype of *Jasione montana* var. *dentata*, designated here (G0032953).

**Figure 5 plants-13-00050-f005:**
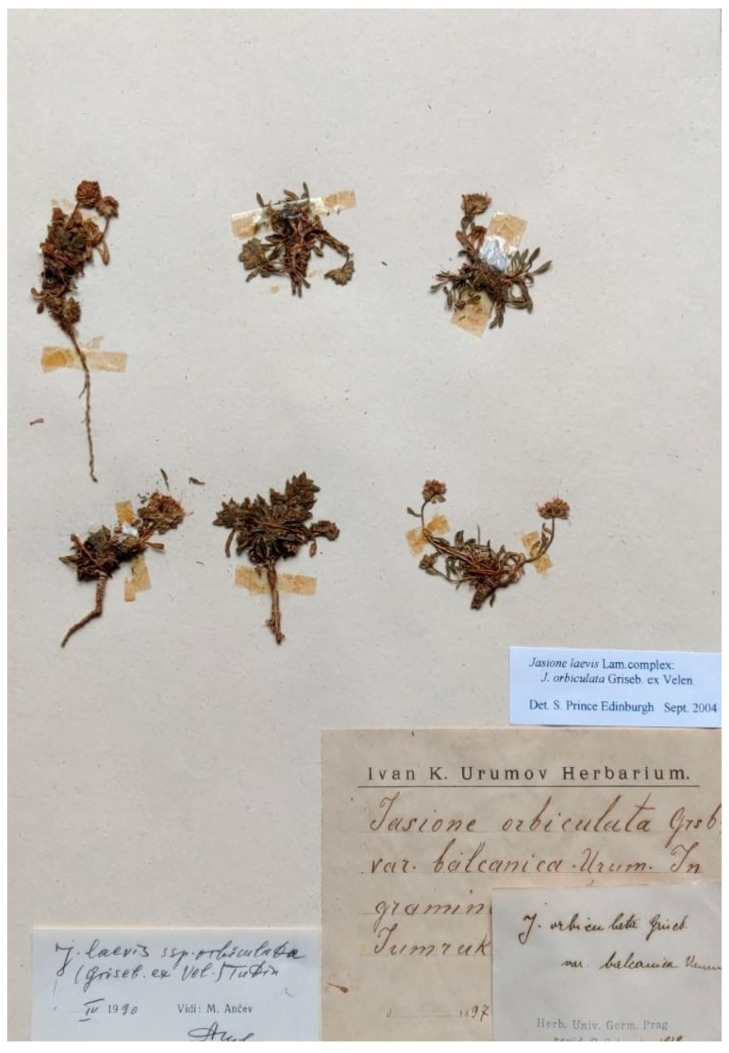
Lectotype of *Jasione orbiculata* var. *balcanica*, designated here (SOM75664).

**Figure 6 plants-13-00050-f006:**
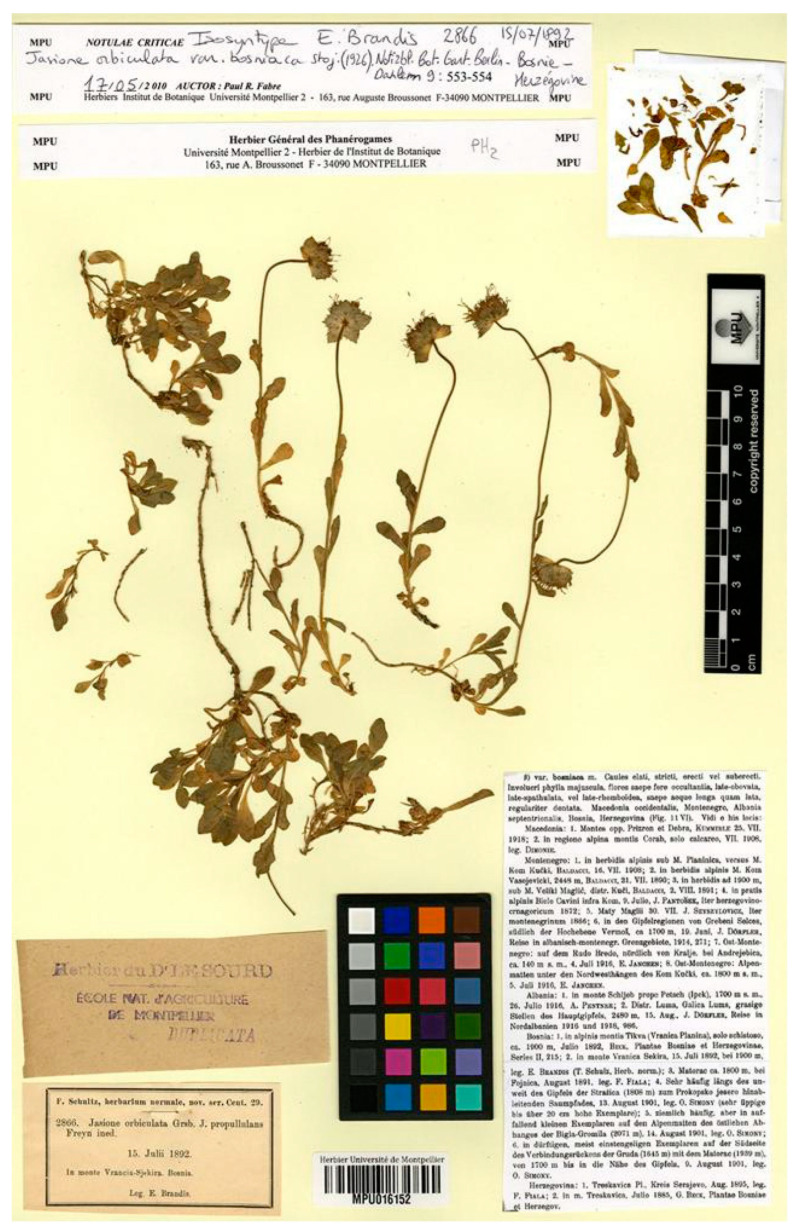
Lectotype of *Jasione orbiculata* var. *bosniaca*, designated here (MPU016152).

**Figure 7 plants-13-00050-f007:**
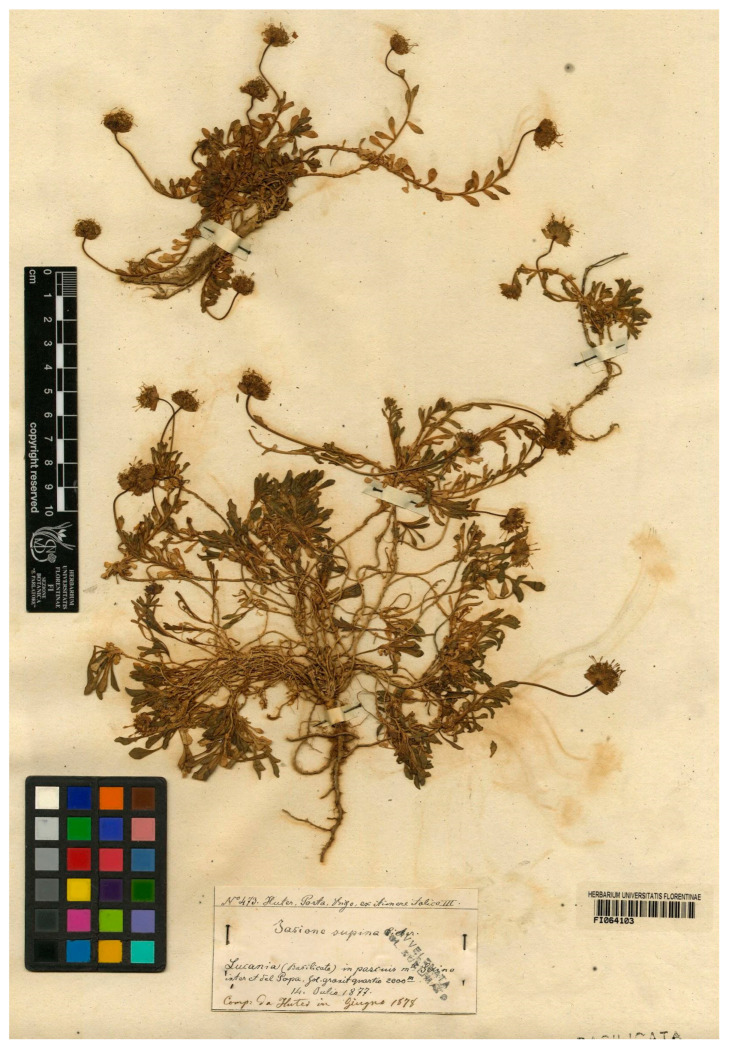
Lectotype of *Jasione orbiculata* var. *italica*, designated here (FI064103).

**Figure 8 plants-13-00050-f008:**
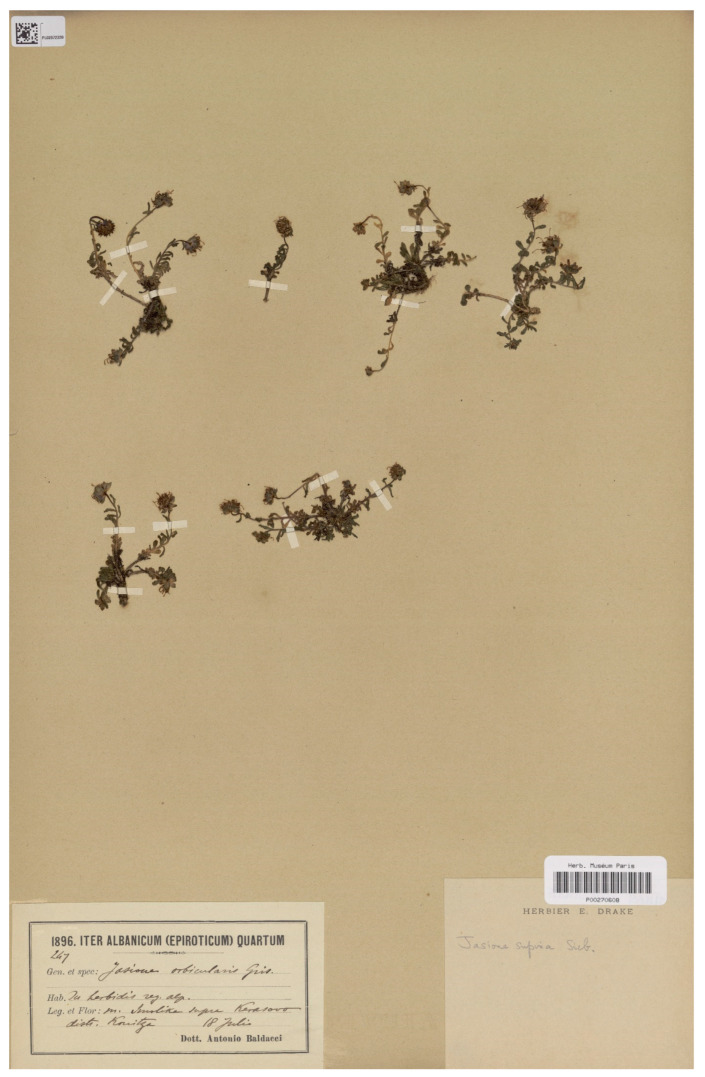
Lectotype of *Jasione orbiculata* var. *supinoides*, designated here (P00270608).

**Figure 9 plants-13-00050-f009:**
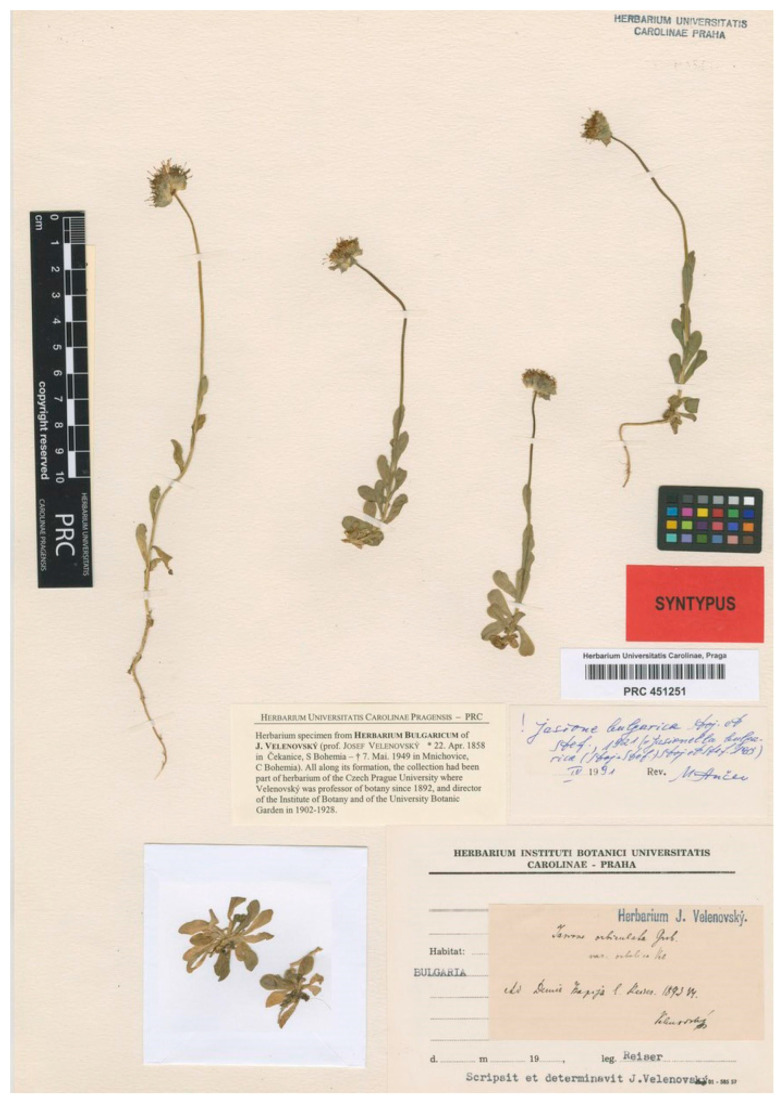
Lectotype of *Jasione orbiculata* var. *orbelica*, designated here (PRC451251).

**Figure 10 plants-13-00050-f010:**
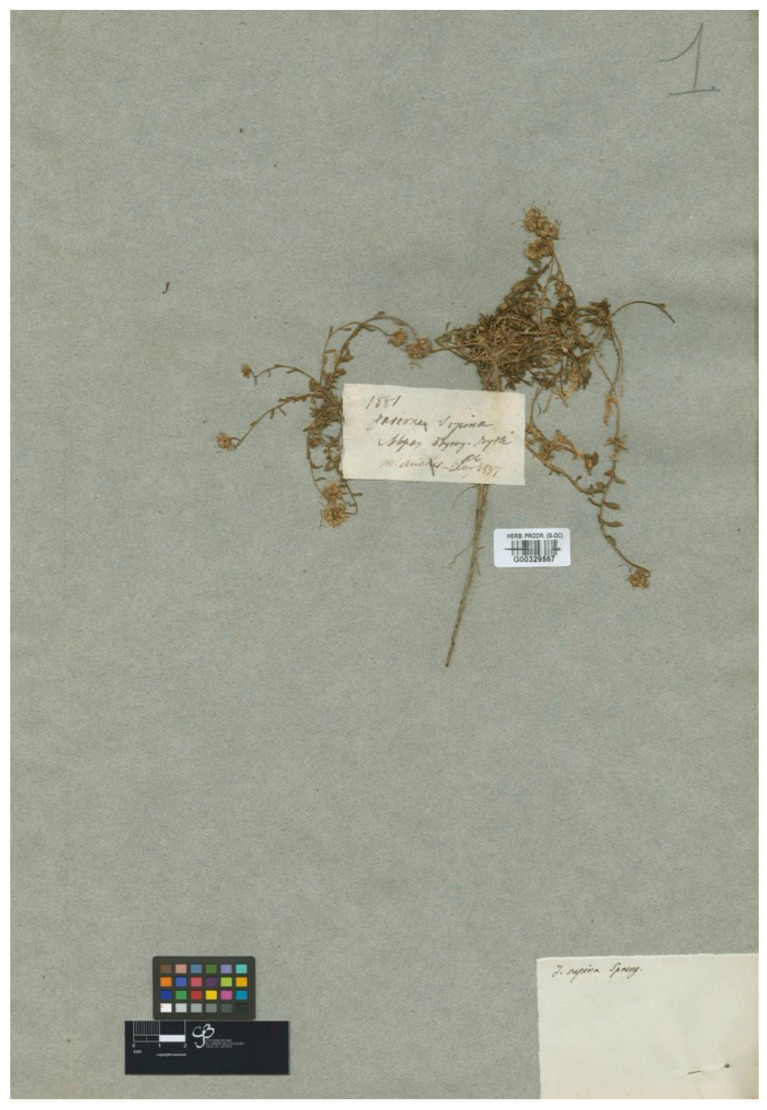
Lectotype of *Jasione supina*, designated here (G00329587).

**Figure 11 plants-13-00050-f011:**
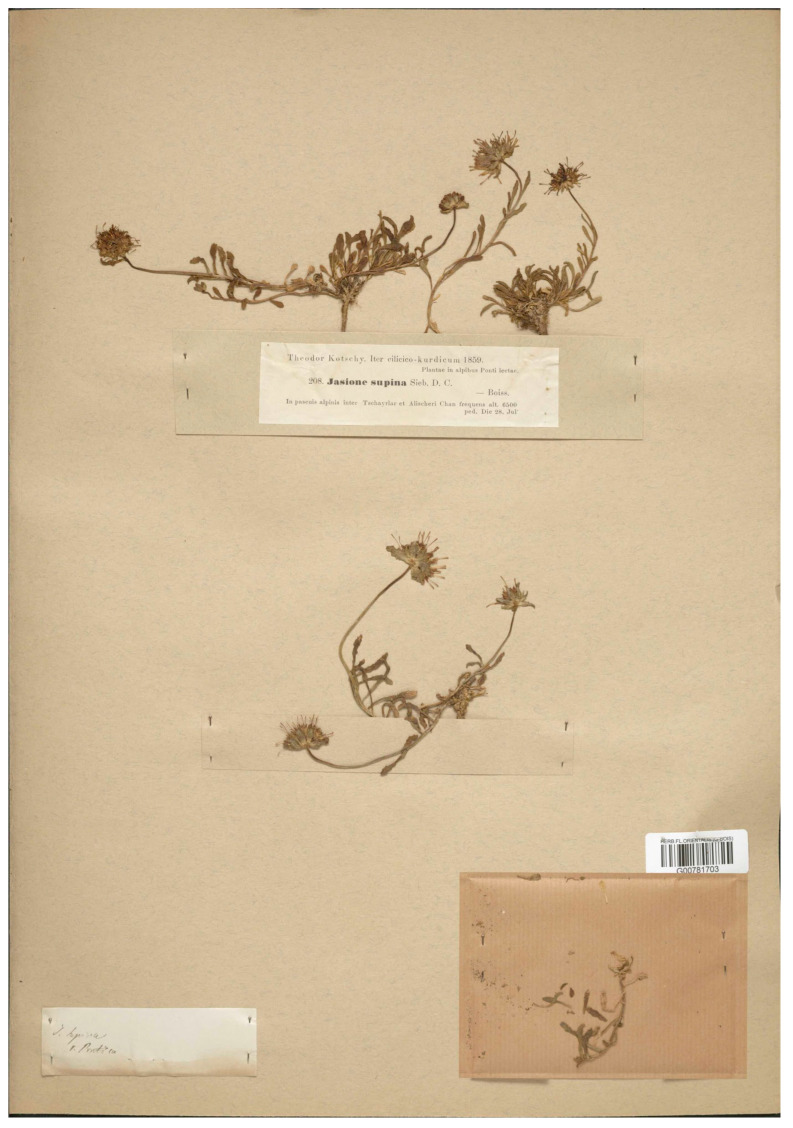
Lectotype of *Jasione supina* subsp. *pontica*, designated here (G00781703).

**Figure 12 plants-13-00050-f012:**
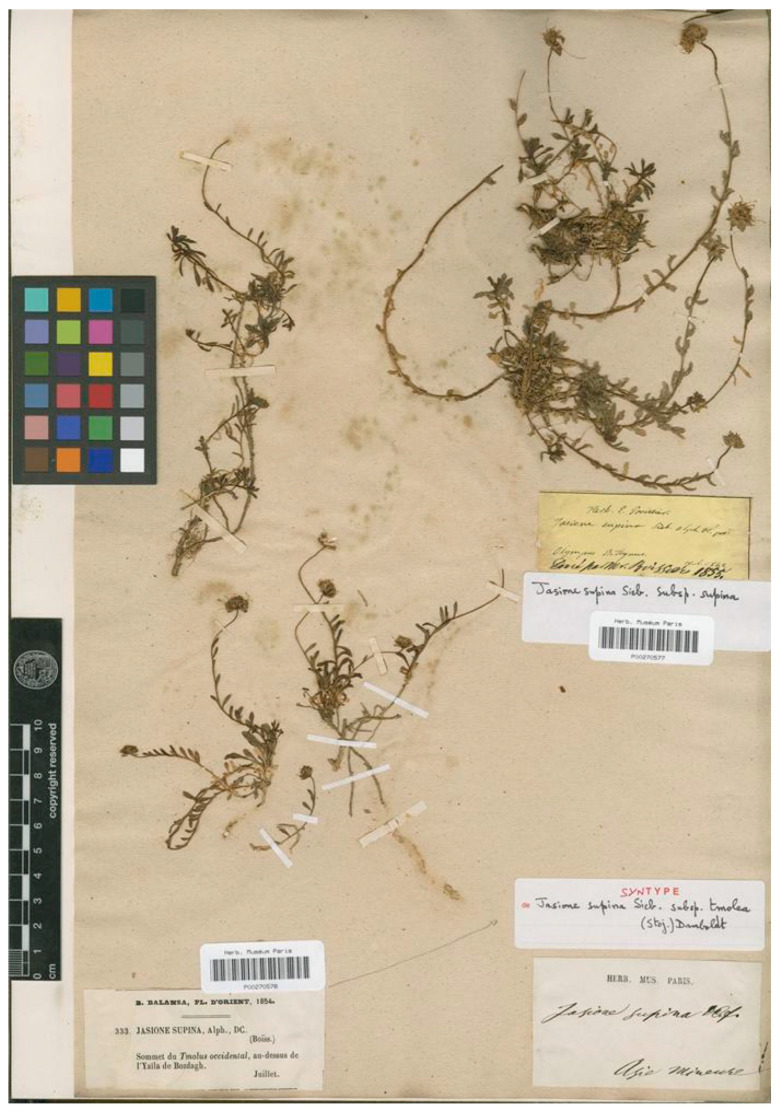
Lectotype of *Jasione supina* subsp. *tmolea*, designated here (P00270580).

**Table 1 plants-13-00050-t001:** Names currently accepted in general treatments (e.g., [4]) are ordered alphabetically in the first column and marked in bold, with basionyms indicated. The second column lists their heterotypic synonyms following current general treatments. Names underlined are those that, in our view, merit taxonomic recognition at the current state of knowledge of ongoing phylogenetic studies and should be accepted.

Accepted Name in General Checklists	Heterotypic Synonym(s)
***Jasione bulgarica* Stoj and Stef.** ≡ *Jasionella bulgarica* (Stoj. and Stef.)	= *Jasione orbiculata* var. *orbelica* Velen.
***Jasione heldreichii *Boiss. and Orph.** ≡ *Jasione montana* L. f. *heldreichii*	= *Jasione glabra* Velen. = *Jasione heldreichii* var. *microcephala* Velen. = *Jasione heldreichii* var. *papillosa* J. Parn. = *Jasione montana* var. *dentata* f. *microcephala* (Velen.) Stoj and Stef. = *Jasione jankae* Neilr. = *Jasione montana* var. *dentata* DC.
** * Jasione idaea * Stoj. **	
** * Jasione * ** ** * orbiculata * ** ** Griseb. Ex Velen. **	= *Jasione orbiculata* var. *balcanica* Urum. = *Jasione orbiculata* var. *bosniaca* Stoj. = *Jasione orbiculata* var. *italica* Stoj. = *Jasione orbiculata* var. *supinoides* Stoj.
** * Jasione * ** ** * supina * ** ** Sieber ex Spreng **	= *Jasione supina* var. *hirtula* Stoj.
***Jasione supina* subsp. *akmanii* Damboldt**	
≡ ***Jasione** supina *subsp. *pontica* (Boiss.) Damboldt** *Jasione supina* var. *pontica* Boiss. (basionym)	= *Jasione pontica* var. *microcephala* Freyn in Bornmüller
≡ ***Jasione supina* subsp. *tmolea* (Stoj.) Damboldt** *Jasione tmolea* Stoj. (basionym)	

***Jasione* L.**, Species plantarum: 928–929. Stockholm, Laurentius Salvius (1753). Type: *J. montana* L. ≡ *Ovilla* Adans., Fam. Pl. 2: 134 (1763). = *Jasionella* Stoj. and Stef., InФл. Бълг. изд. (Flora Bulgarica) 2: 986 (1933). = *Urumovia* Stef., God. Sfiisk. Univ. Agron.-Lesoved. Fak. 14: 103 (1936).

## Data Availability

Data is contained within the article.

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
