# Peer review of "Nomenclature and Typification of the Jasione L. (Campanulaceae) Groups from the Eastern Mediterranean Basin"

_plants, 2023, doi:10.3390/plants13010050_

Round 1
Reviewer 1 Report
Comments and Suggestions for Authors
The authors' proposal is good. However, the reviews are generally confusing. Therefore, numerous questions from a nomenclatural point of view still need to be clarified. Such controversies sometimes reflect the similarities and sometimes the striking morphological differences associated with the specimens studied. Considering what was presented by the authors, the phylogenetic study of Jasione is required and will probably culminate in the publication of the taxonomic review of the genus and, thus, a series of developments from the nomenclatural point of view will be proposed in order to promote the stability of the names associated with Jasione. Considering the reasons mentioned above, I am unfavorable to publishing this work in its current format.

Author Response
We would like to thank the reviewer for his comments and revision, which we have followed in all cases and which we believe have clearly improved the manuscript. As an example, the text in Material and methods sections was enlarged to satisfy the suggestions. A reference to JSTOR international plant platform and other indications regarding the selected herbaria were originally omitted for the sake of conciseness in an already long manuscript. Nevertheless, we agree with reviewer 1 that the way how the herbaria were consulted and the reference to JSTOR platform should be included and find that the material and methods section has been improved thanks to this suggestion.
Regarding the general comment, we agree that a phylogenetic analysis of genus Jasione will be very interesting not only to clarify the evolutionary relationships within the genus but to support or discard a number of taxonomic names. We are working on it and precisely for that reason we think that a thorough nomenclatural review in the so far neglected names of the Balkan and Anatolian regions is an important step to address future biosystematics studies based on a molecular framework. This eastern region of the Mediterranean basin is one of the main evolutionary centres of the genus, what is related to the large number of names that have been proposed. In our opinion, this nomenclatural work is therefore long overdue. A precise connection between each name and its type (and therefore a name and a local population) will offer certainty to decide whether new taxa have to be described or new combinations have to be addressed, in this case on the basis of a clear and well typified basionym.
Reviewer 2 Report
Comments and Suggestions for Authors
the article is very long to be reread in the short time available, I was only able to give a formal review without going into detail. Several things need to be fixed as indicated in the text. The authors propose an unnecessary epitype. The authors propose a neolectotype, this category does not exist, it must be corrected to neotype.

Only minor review is necessary
Author Response
We would like to thank the reviewer for his/her comments and revision, which we have followed in all cases and which we believe have clearly improved the manuscript.
Regarding the most important comments, we have clarified the abstract including an explicit reference to the Balkans and Anatolia, we have changed our view about the lectotype of Jasione bulgarica and accepted the reviewer suggestion and eliminated the epitype for this name, and corrected the neotype issue of Jasione supina.
Reviewer 3 Report
Comments and Suggestions for Authors
This study is devoted to the Nomenclature and typification of the Jasione L. (Campanulaceae) groups from the eastern Mediterranean basin. Although important nomenclatural summaries have been provided for Jasione in areas of the western Mediterranean region. (North African populations, and the Iberian Peninsula) on the Balkan and Anatolian peninsulas it remains neglected for almost a century. The research is carefully provided and presents valuable information and updates. Listed are the names currently accepted in general treatments. The taxonomic status and typification of all published names are discussed. A sufficient number of herbarium specimens are studied and important materials are photographed as illustrations of the publication.
Author Response
The authors would like to acknowledge the review and general comments of reviewer 3.
Round 2
Reviewer 1 Report
Comments and Suggestions for Authors
I think that the paper is ready for publication.
Author Response
Thank very much for your helpful review.
Reviewer 2 Report
Comments and Suggestions for Authors
After typification the authors should indicate their taxonomical proposal (accepted, synonym) for each of the taxa considered. This taxonomic scheme can be summarized in a table. There are many small imperfections (commas, periods) in the text. I have indicated the most obvious ones but all the text should be double-checked. I suggest removing "in" before the indication of name references because sometimes it has been used before the reference of books, sometimes before the reference of journals, sometimes it has not been used.

the English used is sufficiently clear, only minor revision is suggested.
Author Response
Thank you very much for your comments of the second-round review.
Regarding your indication of including a taxonomic proposal in a table we have opted to modify the Table 1 as a compromise solution, a change that we hope could be enough for the publication of the manuscript. Thus, we would prefer to maintain as accepted names, and list their synonyms, those names that are currently accepted in general checklists. Nevertheless, following your suggestion, we now indicate which names would deserve recognition in our view, and therefore should be accepted by a new taxonomic proposal for the genus. However, we think that this taxonomic treatment has to be addressed after the publication of the phylogenetic studies that we are now conducting. Based on those results, the updated taxomomic proposal would imply new combinations. This forthcoming taxonomic proposal would be supported by both the phylogeny and the present work, as it stablish a precise connection between the names (basionyms of future combinations) and their types.
In any case, and trying to follow your suggestion as much as possible, we have modified the table 1 and changed its caption to “Table 1. Names currently accepted in general treatments (e.g., [4]) are ordered alphabetically in the first column and marked in bold, with basionyms indicated. The second column lists their heterotypic synonyms following current general treatments. Names underlined are those that in our view merit taxonomic recognition at the current state of knowledge of ongoing phylogenetic studies and should be accepted.”
We also have inserted a sentence in the text of the manuscript: “Notwithstanding, as a prelude to a more formal taxonomic treatment informed by phylogenetic relationships, Table 1 underlines those names that in the opinion of the current authors would merit taxonomic recognition, irrespective of whether future systematic works may propose for them new combinations of their taxonomic rank.”
We would like to thank you for your thorough review which allowed us to correct all the small imperfections detected by you, and others that we have found after double-checking. We have also removed all the “in” as you suggested.
Round 3
Reviewer 1 Report
Comments and Suggestions for Authors
Dear authors:
- On the page 4, replace please Jasione orbiculata Griseb. Ex Velen. Jasione orbiculata Griseb. ex Velen.
- From the line 371 (until the line 380) replace 13rbiculate with orbiculate
- From the line 441 (until the line 463) to do changes in the name of taxon
- From the line 475 (until the line 485) to do changes in the name of taxon